# 5-Years Analysis of Effectivity and Toxicity of Ultra-Hypofractionated Proton Radiotherapy in the Treatment of Low- and Intermediate-Risk Prostate Cancer—A Retrospective Analysis

**DOI:** 10.3390/cancers15184571

**Published:** 2023-09-15

**Authors:** Jiri Kubeš, Silvia Sláviková, Pavel Vítek, Alexandra Haas, Barbora Ondrová, Kateřina Dedečková, Michal Andrlík, Martin Domanský, Kateřina Jiránková, Veronika Schlencová, Anh Harazimová, Barbora Turková, Tomáš Doležal, Sarah Falah Abass Al-Hamami, Vladimír Vondráček

**Affiliations:** 1Proton Therapy Center Czech, Budínova 1a, 180 00 Prague, Czech Republic; jiri.kubes@ptc.cz (J.K.); silvia.slavikova@ptc.cz (S.S.); pavel.vitek@ptc.cz (P.V.); alexandra.haas@ptc.cz (A.H.); barbora.ondrova@ptc.cz (B.O.); katerina.dedeckova@ptc.cz (K.D.); m.domansky@seznam.cz (M.D.); jirankova.katerina@seznam.cz (K.J.); veronika.schlencova@gmail.com (V.S.); harazimova.anh@gmail.com (A.H.); sarah.alhamami@ptc.cz (S.F.A.A.-H.); vladimir.vondravek@ptc.cz (V.V.); 2Department of Health Care Disciplines and Population Protection, Faculty of Biomedical Engineering, Czech Technical University Prague, Sítná Square 3105, 272 01 Kladno, Czech Republic; 3Department of Oncology, 1st Faculty of Medicine and General University Hospital, Charles University, Kateřinská 32, 121 08 Praha, Czech Republic; 4Value Outcomes Ltd., Václavská 316/12, 120 00 Praha, Czech Republic; barbora.turkova@valueoutcomes.cz (B.T.); tomas.dolezal@valueoutcomes.cz (T.D.)

**Keywords:** prostate cancer, proton therapy, pencil beam scanning, ultra-hypofractionated radiotherapy, late toxicity

## Abstract

**Simple Summary:**

This retrospective study presents the clinical outcomes of the largest cohort of patients (853 patients) with low-, favorable intermediate-, and unfavorable intermediate-risk prostate cancer treated with ultra-hypofractionated proton beam radiotherapy (36.25 GyE/five fractions). The median follow-up time was 62.7 months. Ultra-hypofractionated proton beam radiotherapy is an effective treatment for low- and favorable intermediate-risk prostate cancer, with long-term bDFS rates comparable to other techniques. It is promising for unfavorable intermediate-risk prostate cancer and has acceptable long-term GI and favorable GU toxicity.

**Abstract:**

Background: We retrospectively analyzed the 5-year biochemical disease-free survival (bDFS) and occurrence of late toxicity in prostate cancer patients treated with pencil beam scanning (PBS) proton radiotherapy. Methodology: In the period from January 2013 to June 2018, 853 patients with prostate cancer were treated with an ultra-hypofractionated schedule (36.25 GyE/five fractions). The mean PSA value was 6.7 (0.7–19.7) µg/L. There were 318 (37.3%), 314 (36.8%), and 221 (25.9%) patients at low (LR), favorable intermediate (F-IR), and unfavorable intermediate risk (U-IR), respectively. Neoadjuvant hormonal therapy was administered to 197 (23.1%) patients, and 7 (0.8%) patients had adjuvant hormonal therapy. The whole group of patients reached median follow-up time at 62.7 months, and their mean age was 64.8 (40.0–85.7) years. The bDFS rates and late toxicity profile were evaluated. Results: Median treatment time was 10 (7–38) days. Estimated 5-year bDFS rates were 96.5%, 93.7%, and 91.2% for low-, favorable intermediate-, and unfavorable intermediate-risk groups, respectively. Cumulative late toxicity (*CTCAE v4*.0) of G2+ was as follows: gastrointestinal (GI)—G2: 9.1%; G3: 0.5%; genitourinary (GU)—G2: 4.3%, and no G3 toxicity was observed. PSA relapse was observed in 58 (6.8%) patients: 16 local, 22 lymph node, 4 bone recurrences, and 10 combined sites of relapse were detected. Throughout the follow-up period, 40 patients (4.7%) died, though none due to prostate cancer. Conclusion: Ultra-hypofractionated proton beam radiotherapy is an effective treatment for low- and favorable intermediate-risk prostate cancer, with long-term bDFS rates comparable to other techniques. It is promising for unfavorable intermediate-risk prostate cancer and has acceptable long-term GI and favorable GU toxicity.

## 1. Introduction

Proton radiotherapy has been used in the treatment of prostate cancer for many decades, and data have repeatedly confirmed its high effectiveness and safety. The majority of published results are for normofractionated radiotherapy, owing to historical practice and a general lack of previously published data for ultra-hypofractionated radiotherapy. After the publication of data for ultra-hypofractionated schedules for photon radiotherapy, and after a subsequent comparison of dosimetric parameters for photon and proton radiotherapy in ultra-hypofractionated modes was reported, these schedules also entered the field of proton radiotherapy [1]. In general, the advantages offered by ultra-hypofractionated regimens include an increase in the therapeutic window due to favorable α/β ratios for prostate cancer cells and the surrounding bladder and rectum, significant improvement in patient comfort during treatment, and reduced treatment costs for the healthcare payer. Disadvantages include the higher impact of errors, particularly in geography, when applying a smaller number of high-dose fractions. We have been using the ultra-hypofractionated regimen in the treatment of low- and intermediate-risk prostate cancer since 2013, and the technique and results of the pilot study have been published previously [2]. We have now performed a retrospective analysis on a larger number of patients in order to verify the previously published results.

## 2. Materials and Methods

Between February 2013 and June 2018, we treated a total of 883 patients with a regimen of 36.25 GyE in 5 fractions over 10 days. A total of 853 patients met the inclusion criteria for the analysis. The criteria included patients with low- or intermediate-risk prostate cancer (according to NCCN classification version 1.2023), with an initial PSA < 15 ug/L and without previous prostate surgery (including transurethral resection), meeting the volume criterion of a planning target volume (PTV) size up to 150 cm^3^ and with a follow-up time of longer than 6 months. All patients had prostate and pelvic MRI scanning as a staging examination. Patients with Gleason score 7 (4 + 3) or patients with MRI-suspected lymph nodes underwent F-choline or Ga-PSMA PET/CT to exclude the extra-prostatic disease. MRI was used for the prostate delineation during the planning procedure. PTV was defined as prostate + 5 mm margin for the LR group. For F-IR and U-IR patients, CTV was first defined as prostate + 5 mm (excluding rectum and bladder), and then PTV was defined as CTV + 5 mm.

The treatment procedures used have been described previously [3]. Briefly, patients were treated with a regimen of 5 × 7.25 GyE every other day for a scheduled total treatment time of 9–10 days. The application of gold fiducial markers was performed in all patients, rectal spacers were permitted, and rectal balloons were not used. The application of neoadjuvant hormonal treatment was at the discretion of the attending physician; however, it was not recommended for low-risk cancer, optional for favorable intermediate-risk cancer, and recommended for unfavorable intermediate-risk cancer. Adjuvant hormonal treatment was not recommended for any of the mentioned risk-groups, and if it was administered, it was at the discretion of the treating urologist. The study was approved by the institutional ethics committee and all patients signed informed consent, which included agreement to data collection and publication.

### 2.1. The Follow-Up Period: Evaluation of Efficacy and Toxicity

Follow-up time was defined as the time from the last fraction of radiotherapy to the last follow-up visit or to the death of the patient. Follow-up visits consisted of clinical examination, PSA measurement, and evaluation of toxicity. Relapse was considered either due to a relapse in PSA according to the Phoenix criteria (PSA nadir plus 2 ng/mL) [4] or evidence of metastasis according to imaging methods. Pelvic MRI and/or PET/CT scan with F-choline for recurrence localization were performed in the case of biochemical failure. Biochemical disease-free survival was defined as time from last fraction of radiotherapy to PSA relapse, last follow-up visit, or death. Toxicity was assessed according to the Common Terminology Criteria for Adverse Events (CTCAE) version 4.0. Any medication or argon laser coagulation taken/conducted 3 months or later after the end of treatment were considered to be instances of grade (G)-2 late GI toxicity. The time to biochemical relapse was defined as the time from the end of radiotherapy to the time when PSA relapse was detected. The time to late toxicity was defined as the time from the end of radiation to the occurrence of toxicity. Demographic and treatment parameters are shown in Table 1.

### 2.2. Statistical Analysis

Continuous data were used as the mean, standard deviation, median and range (minimum and maximum), and descriptive statistics with which to evaluate categorical data as frequencies with percentages. Kaplan–Meier survival curves were estimated for biochemical disease-free survival (bDFS), overall survival (OS), and cumulative incidence of late GI/GU toxicity, and these were compared with the log-rank test. A Cox proportional hazards model was used to analyze the impact of Gleason score, baseline PSA, age, T-stage, neoadjuvant hormonal therapy, and total radiotherapy time on bDFS. The significance level was set at 5%. Statistical analysis was performed using R software 4.2.3 [5].

## 3. Results

In the institutional database, we identified a total of 883 patients treated in the mentioned period. A total of 30 patients were excluded due to insufficient follow-up time (they did not continue in follow-up in our institution for longer than 6 months); therefore, 853 patients were analyzed. The median follow-up (FU) time was 62.7 months (6.4–117.3 months), with 666 (78.1%) patients reaching an FU time of longer than 48 months and 454 (53.2%) patients reaching an FU time of longer than 60 months. The representation of individual risk groups according to the NCCN guidelines version 1.2023 was as follows: low risk (LR): 318 (37.3%); favorable-intermediate risk (F-IR): 314 (36.8%); and unfavorable-intermediate risk (U-IR): 221 (25.9%). All patients were treated with a dose of 36.25 GyE in five fractions over 9–10 days. The only difference in radiotherapy between LR and IR patients was a 0.5 cm expansion of GTV for the CTV, which was used for F-IR and U-IR patients and not for LR patients. The median and mean duration of radiotherapy was 10 days (7–38 days, SD 1.9). Neoadjuvant hormonal treatment was applied in 17 (5.3%), 80 (25.5%), and 100 (45.2%) LR, F-IR, and U-IR patients, respectively. Adjuvant hormonal treatment was not indicated in any patient; however, in 7 (0.8%) patients, it was applied at the decision of the attending urologist.

### 3.1. Biochemical Control

Biochemical relapse was detected in a total of 58 (6.8%) patients. The 5-year biochemical disease-free survival (bDFS) was 94.1% for the entire cohort, and 96.5%, 93.7%, and 91.2% for the LR, F-IR, and U-IR groups, respectively. The Kaplan–Meier curve for bDFS is shown in Figure 1.

The localization of relapse was as follows: local in 16 (1.8%) patients, nodal in 22 (2.6%) patients, bone in 4 (0.5%) patients, combined in 10 (1.2%) patients, and not localized in 6 (0.7%) patients. The location of PSA relapse is shown in Figure 2.

Forty (4.7%) patients died in total, with none of the deaths attributed to prostate cancer. The 5-year OS was 95.6% for the entire cohort, and 96.2%, 94.8%, and 95.9% for the LR, F-IR, and U-IR groups, respectively. The Kaplan–Meier survival curve for OS is shown in Figure 3.

### 3.2. Late Toxicity

Late toxicity was graded according to *CTCAE v4*.0. The cumulative 5-year gastrointestinal (GI) toxicity of G1, G2, and G3 was 17.5%, 9.1%, and 0.5%, respectively. The incidence of G2+ GI toxicity during the whole follow-up period for individual subgroups was 6.0%, 9.2%, and 14.4% for the LR, F-IR, and U-IM groups, respectively. The cumulative 5-year genitourinary (GU) toxicity of G1 and G2 was 29.4% and 4.3%, respectively. GU toxicity of G3 and higher was not noted. The incidence of G2+ GU toxicity during the whole follow-up period for individual subgroups was 3.8%, 4.1%, and 4.6% for the LR, F-IR, and U-IM groups, respectively. The incidence of late GI and GU toxicity is shown in Figure 4.

### 3.3. Factors Influencing Relapse-Free Survival

An analysis of potential factors affecting biochemical control using the Cox proportional-hazards model was performed. T classification, age or duration of radiotherapy did not influence bDFS. The application of neoadjuvant hormonal therapy also did not affect bDFS, even for the U-IR group of patients. The initial PSA value (<10 vs. >10 µg/L; *p* = 0.008) and Gleason score (<7 vs. 7; *p* = 0.021) were found to have an effect on bDFS (Table 2). The influence of individual factors was graphically represented by the Kaplan–Meier survival curves. Although risk group was identified as a significant factor of bDFS by the log-rank test, it is no longer significant in the Cox model due to correlation with initial PSA value and Gleason score.

## 4. Discussion

Proton radiotherapy for prostate cancer is an accepted treatment modality with sufficient published data from non-randomized trials. Its potential advantages over advanced photon radiotherapy techniques are still debated. The higher cost of proton radiotherapy and its limited availability are considered significant disadvantages. We have conducted the largest study to date describing the long-term results of ultra-hypofractionated proton radiotherapy for prostate cancer, where the use of a small number of therapeutic fractions offers a possible solution to both of the aforementioned disadvantages. This analysis is an extension of previously published work based on the analysis of a smaller number of patients with the aim of verifying the results in a sufficiently large cohort of patients.

### 4.1. Comparison with Ultra-Hypofractionated Photon Radiotherapy

Photon radiotherapy with similar regimens is an established method and results have been published. For example, Kishan A.U. et al. [6], with a median follow-up of 6.9 years, the 7-year bDFS for low-, favorable intermediate-, and unfavorable intermediate-risk patients was 95.5%, 91.4%, and 85.1%, respectively, with G3+ late toxicity in 2.4% of patients for the GU system and 0.4% for GI. Katz and Kang described results of 477 patients with a median follow-up of 72 months. The 7-year bDFS was 95.6% and 89.6% for the low- and intermediate-risk groups, respectively. Late toxicity was low, with grade-2 rectal and urinary toxicity of 4% and 9.1%, respectively, as well as a grade-3 urinary toxicity of 1.7% [7]. Fuller et al. recently published 10-year results for ultra-hypofractionated photon radiotherapy using 38 Gy in four fractions. They reported 10-year freedom from biochemical recurrence results of 100%, 84.3%, and 68.4% for LR, F-IR, and U-IR, respectively. The cumulative incidence of grade-2 GU toxicity was 16.3% at 5 years and increased to 19.2% at 10 years. Incidence of grade-2 and -3+ GI toxicity was 4.1% and 0%, respectively [8]. The results for our group of proton-treated patients are favorable in comparison, especially for the unfavorable intermediate-risk patients.

### 4.2. Comparison with Normofractionated or Mild Accelerated Proton Radiotherapy

Proton radiotherapy is used mostly in normofractionated or slightly hypofractionated regimens. One of the more extensive studies was published by Takagi et al. [9]. Using a regimen of 74 GyE/37 fractions and a median follow-up of 84 months, the 5-year freedom from biochemical relapse results rates were 100%, 99%, 93%, and 90% for the very low-, low-, favorable intermediate-, and unfavorable intermediate-risk groups, respectively. The 5-year rates of G2+ late genitourinary and gastrointestinal toxicity were 2.2% and 4.0%, respectively.

The results of slightly accelerated proton beam radiotherapy in the treatment of prostate cancer were published by Henderson H et al. [10]. They used a schedule of 70 GyE in 28 fractions. In 215 patients with a median follow-up of 5.2 years, the 5-year bDFS in low- and intermediate-risk patients was 98.3% and 92.7%, respectively. G3+ toxicity for GU and GI was 0.5% and 1.7%, respectively [10].

Grewal et al. described the 4-year results of the regimen 70 GyE in 28 fractions in a group of 184 men with a median follow-up of 49.2 months. The 4-year bDFS rates were 94.4%, 92.5%, and 93.8% for the LR, F-IR, and U-IR groups, respectively [11]. The incidence of late G2+ GU and GI toxicity was 7.6% and 13.6%, respectively. The authors described one patient with late G3 GI toxicity. Slater et al. treated 146 men with low-risk prostate cancer with a regimen of 60 GyE/20 fractions over 4 weeks [12]. With a median follow-up of 42 months, the 3-year bDFS was 99.3%. There was no G3 GI toxicity and one case of G3 GU toxicity in the cohort.

### 4.3. Comparison with Proton Ultra-Hypofractionated Radiotherapy

There are only a few publications describing the results of proton ultra-hypofractionation in prostate cancer. Vargas et al. present a comparison of toxicity and quality of life for extremely hypofractionated and normo-fractionated proton radiotherapy [13]. When comparing the schedules of 38 GyE in 5 fractions and 79.2 GyE in 44 fractions, they had low toxicity in both arms and a temporarily worse genitourinary score in the ultra-hypofractionated arm. Ha et al. published the results of a phase-2 study (82 patients) treated with a mildly accelerated or ultra-hypofractionated regimen [14]. There were 30 patients of all risk groups in the ultra-hypofractionated group, which used a regime of either 35 GyE/five fractions/2.5 weeks or 35 GyE/five fractions/4 weeks. At a median follow-up of 7.5 years, the 7-year bDFS was 57.1% and 42.9% for the low- and intermediate-risk groups, respectively, under the ultra-hypofractionated regime. Slightly accelerated proton radiotherapy had significantly better results and late GI and GU toxicities did not differ between the groups. Compared to the above-mentioned publications, our patient cohort showed favorable outcomes, especially for the U-IR group. GI toxicity is comparable to the previously published data, while GU toxicity is favorable in our study compared to those using photon techniques.

### 4.4. Pattern of Relapse

Location of PSA relapses were reported Miszczyk by et al. after utilizing a 36.25 Gy/five fraction schedule for 650 patients treated with CyberKnife. In their group, there were 23 (3.5%) patients with isolated local recurrences [15]. Henderson et al. described two (0.9%) local recurrences in a group of 215 patients treated with accelerated proton radiotherapy [10]. Bao et al. described a pattern of local relapses in the group of 166 patients with favorable risk prostate cancer treated with proton beam radiotherapy with 79.2 GyE in 44 fractions. With a median follow-up time of 8.3 years, they found 13 (7.8%), 8 (4.8%), and 2 (1.2%) patient(s) with biochemical failure, local failure, and regional failure, respectively. None of their patients experienced distant relapse [16]. The 1.8% occurrence of local relapses in our study does not deviate from the described rates of local relapses.

### 4.5. Comparison with Previous Results from Our Institution

When comparing the results of this study with those of our previous study, it is clear that for LR and F-IR patients, the results are very similar, even when the set is expanded (96.5% vs. 96.9% and 93.7% vs. 91.7%). Late GU and GI toxicity are also similar (4.3% vs. 5% and 9.6% vs. 7.6%). A significant improvement in bDFS for U-IR patients is evident (91.2% vs. 83.5%). The probable explanation for this improvement is the introduction of a PET/CT examination with F-choline [17] or PSMA [18] for patients with U-IR prostate cancer in mandatory staging before initiating therapy.

### 4.6. Strengths and Weaknesses

To the best of our knowledge, this is the largest published cohort of patients treated with ultra-hypofractionated proton radiotherapy using the PBS technique. The strength of the work is mainly the homogeneity of the group—all patients were treated according to the same procedures, with the same dose and essentially the same planning approach. An obvious disadvantage of the work is its retrospective nature, as well as possible selection bias due to the potentially higher socioeconomic status of patients coming for proton radiotherapy.

## 5. Conclusions

Ultra-hypofractionated proton radiotherapy using PBS is highly effective in the treatment of low- and intermediate-risk prostate cancer. We confirmed our previously published data regarding the favorable profile of late gastrointestinal and genitourinary toxicity. According to our experience, this fractionation regimen increases the capacity of the proton treatment facility and could improve acceptance of this treatment by healthcare providers by reducing treatment costs.

## Figures and Tables

**Figure 1 cancers-15-04571-f001:**
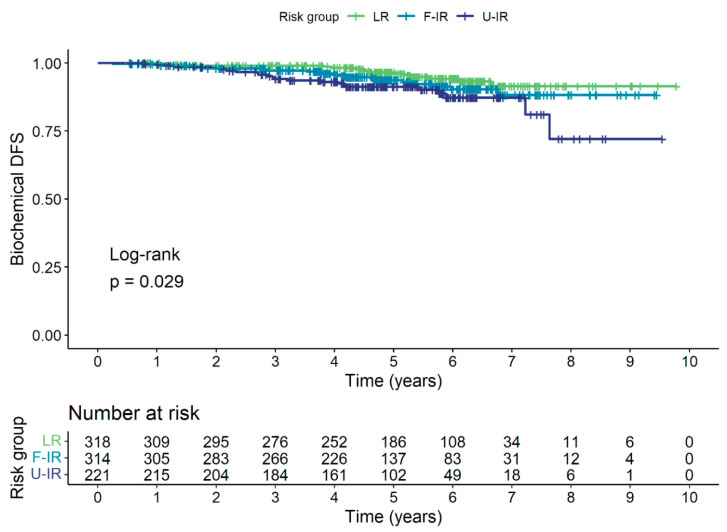
Kaplan–Meier curve for bDFS for low-risk (LR), favorable intermediate-risk (F-IR), and unfavorable intermediate-risk (U-IR) groups.

**Figure 2 cancers-15-04571-f002:**
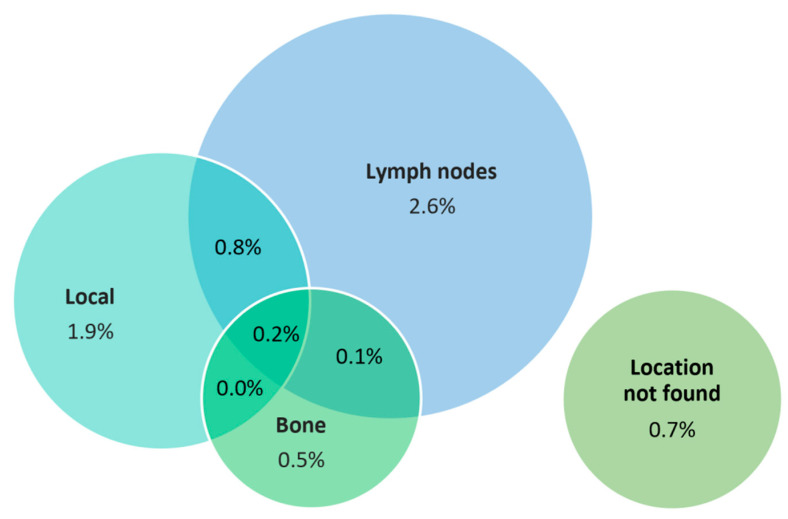
Location of PSA relapse.

**Figure 3 cancers-15-04571-f003:**
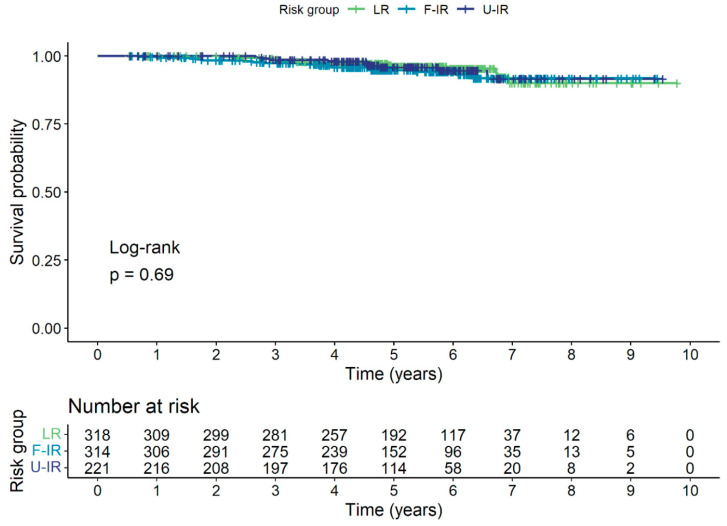
Kaplan–Meier survival curve for OS for low-risk (LR), favorable intermediate-risk (F-IR), and unfavorable intermediate-risk (U-IR) groups.

**Figure 4 cancers-15-04571-f004:**
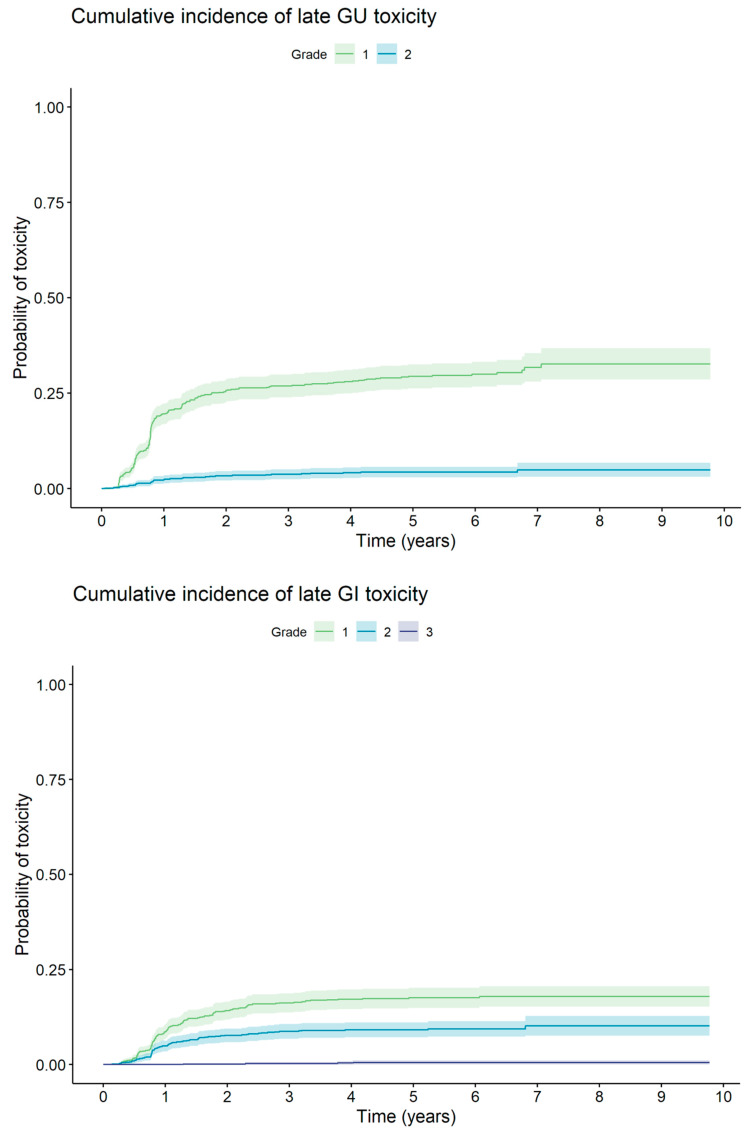
Cumulative incidence of late GI and GU toxicity.

**Table 1 cancers-15-04571-t001:** Demographic and treatment parameters of patient group.

	Number	
N	853	100.00%
Age (years)	65.0 (median)	40.0–85.7 (range)
Adenocarcinoma	853	100.00%
Risk group 1 *	318	37.30%
Risk group 2a *	314	36.80%
Risk group 2b *	221	25.90%
T stage		
T1a-c	372	43.60%
T2a-b	237	27.80%
T2c	244	28.60%
Gleason score		
7 (3 + 4)	184	21.50%
7 (4 + 3)	54	6.30%
7 (NA)	5	0.60%
<7	599	70.20%
Not specified	11	1.30%
PSA		
<10	744	87.20%
10–20	109	12.80%
Neoadjuvant hormonal treatment YES/NO **	197/656	23.1/76.9%
Adjuvant hormonal treatment YES/NO **	7/846	0.8/99.2%
Radiotherapy—total dose (GyE)	36.25	100%
Radiotherapy—overall treatment time	10/median	7–38/range

* Risk group (according to the National Comprehensive Cancer Network (NCCN)). ** Neo/Adjuvant hormonal treatment (androgen therapy—LHRH analogue, androgen).

**Table 2 cancers-15-04571-t002:** Hazard ratios for variables influencing bDFS (Cox model).

Variable	Value	No. of Patients	HR	95% CI	*p*
Initial PSA, ng/mL	<10	744	1.000	(ref)	-
	10–20	109	2.37	1.26–4.46	0.008 *
Gleason score	<7	599	1.000	(ref)	-
	7	243	2.00	1.11–3.59	0.021 *
Age, y	<65	414	1.000	(ref)	-
	65+	439	1.25	0.74–2.11	0.398
Duration of RT, d	<10	264	1.000	(ref)	-
	10+	589	1.30	0.73–2.31	0.367
Neoadjuvant hormonal treatment	No	656	1.000	(ref)	-
	Yes	197	0.88	0.46–1.68	0.697
T stage	T1a-c	372	1.000	(ref)	-
	T2a-b	237	1.01	0.54–1.88	0.983
	T2c	244	0.91	0.47–1.77	0.787

Abbreviations: HR = hazard ratio; RT = radiation therapy. * Statistically significant.

## Data Availability

Anonymized-patient data for this study are available upon request from the corresponding author.

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
