# Peer review of "5-Years Analysis of Effectivity and Toxicity of Ultra-Hypofractionated Proton Radiotherapy in the Treatment of Low- and Intermediate-Risk Prostate Cancer—A Retrospective Analysis"

_cancers, 2023, doi:10.3390/cancers15184571_

Round 1

Reviewer 1 Report

The authors present interesting data on the PCa treatment with ultra-hypofractionated proton therapy with an appropriate follow-up both for oncological and toxicity outcomes. The present work is an update of data of already published data witha  longer follow-up. in my opinion could be accpeted in the present form. 

Author Response

Thank you very much for reviewing our article.

Reviewer 2 Report

I believe that it is necessary to define some aspects of the diagnostic and therapeutic process, such as defining whether the patients were staged with MRI and what were the PTVs used for treatment. I believe that these aspects can improve the usefulness of the article.

Author Response

Thank you very much for reviewing our article.

Author's Reply to the Review Report (Reviewer 2)

Comments and Suggestions for Authors

I believe that it is necessary to define some aspects of the diagnostic and therapeutic process, such as defining whether the patients were staged with MRI and what were the PTVs used for treatment. I believe that these aspects can improve the usefulness of the article.

Staging procedure and PTV margin criteria were added to chapter materials and methods.

Reviewer 3 Report

The strengths of this study lie in its innovative exploration of the effectiveness and safety of Ultrahypofractionated Proton Radiation Therapy in prostate cancer. Currently, there are few published articles on this topic, and compared to those already published, this study has the largest number of cases. Concurrently, the disease grading and toxicity classification in this study rigorously followed established guidelines. Additionally, all patients were treated according to the same procedures, with the same dose and essentially the same planning approach, leading to high homogeneity among the patients. However, the study's limitations are also quite evident. Firstly, it is a retrospective study, which inevitably introduces confounding factors. Moreover, there is a lack of control group, potentially leading to an overestimation of the actual effectiveness.

In addition, I have some questions about this research :

Q1:

It is extremely important to define biochemical disease-free survival(bDFS) in this text.

Q2:

The baseline assessment categorizes the Gleason score into two groups: 7 points and <7 points. However, the International Society of Urological Pathology (ISUP) recommends a more detailed categorization of the Gleason score into 5 grades, wherein Grade II corresponds to 3+4=7 and Grade III corresponds to 4+3=7. This distinction is crucial as prognosis differ between these two subcategories despite both having a Gleason score of 7. Further subdivision should be considered, including subsequent Cox analysis, to account for these variations in prognosis based on the different subtypes of Gleason score 7.

Q3

The exclusion criteria for the 30 patients mentioned in the results section are not explicitly outlined in the article. It is important for the study's transparency and clarity to clearly state the criteria by which these patients were excluded from the research. This would provide a comprehensive understanding of the patient selection process.

Q4

Can cumulative late toxicity be separately listed and compared in different subgroups?

Q5The discussion section should not solely directly quote contents of references. Instead, it should involve the authors' analysis, and synthesis of the references to provide their own insights.

Q6

Compared with previous articles in the same topic, what are the innovation points and advantages of the author's research? It should be discussed in the discussion part.

Q7

If the research design could be visualized as a flowchart, it would enhance the clarity  of the study process.

Q8

Although most of the data results were consistent with those previously published, it showed a significant increase in the biochemical progression-free survival of U-IR patients compared with the previous outcome, and the reasons for this distinction should be discussed.

Author Response

Thank you very much for reviewing our article.

Author's Reply to the Review Report (Reviewer 3)

Comments and Suggestions for Authors

The strengths of this study lie in its innovative exploration of the effectiveness and safety of Ultrahypofractionated Proton Radiation Therapy in prostate cancer. Currently, there are few published articles on this topic, and compared to those already published, this study has the largest number of cases. Concurrently, the disease grading and toxicity classification in this study rigorously followed established guidelines. Additionally, all patients were treated according to the same procedures, with the same dose and essentially the same planning approach, leading to high homogeneity among the patients. However, the study's limitations are also quite evident. Firstly, it is a retrospective study, which inevitably introduces confounding factors. Moreover, there is a lack of control group, potentially leading to an overestimation of the actual effectiveness.

In addition, I have some questions about this research :

Q1:

It is extremely important to define biochemical disease-free survival(bDFS) in this text.

bDFS was defined in section” The follow-up period: evaluation of efficacy and toxicity”.

Q2:

The baseline assessment categorizes the Gleason score into two groups: 7 points and <7 points. However, the International Society of Urological Pathology (ISUP) recommends a more detailed categorization of the Gleason score into 5 grades, wherein Grade II corresponds to 3+4=7 and Grade III corresponds to 4+3=7. This distinction is crucial as prognosis differ between these two subcategories despite both having a Gleason score of 7. Further subdivision should be considered, including subsequent Cox analysis, to account for these variations in prognosis based on the different subtypes of Gleason score 7.

GS 7 (4+3 vs 3+4) patient specification was more clarified in table 1.

Q3

The exclusion criteria for the 30 patients mentioned in the results section are not explicitly outlined in the article. It is important for the study's transparency and clarity to clearly state the criteria by which these patients were excluded from the research. This would provide a comprehensive understanding of the patient selection process.

We have added one explanatory sentence in the” Results” section. These patients themselves declined follow-up visits or could not able be contacted.

Q4

Can cumulative late toxicity be separately listed and compared in different subgroups?

It was added in “Results “section.

Q5The discussion section should not solely directly quote contents of references. Instead, it should involve the authors' analysis, and synthesis of the references to provide their own insights.

We added several references for the reason to more comprehensive discussion in the chapter “Discussion”.

Q6

Compared with previous articles in the same topic, what are the innovation points and advantages of the author's research? It should be discussed in the discussion part.

We tried to explain better the advantage of presented research.

Q7

If the research design could be visualized as a flowchart, it would enhance the clarity  of the study process.

We suppose that this flow chart is not necessary, because it would be too simple.

Q8

Although most of the data results were consistent with those previously published, it showed a significant increase in the biochemical progression-free survival of U-IR patients compared with the previous outcome, and the reasons for this distinction should be discussed.

We believe that the reason has already been stated in the original text - the introduction of PET CT with PSMA or F choline as a mandatory test for U-IR.

Round 2

Reviewer 3 Report

accept.